# Depressive, Anxiety, and Stress Symptoms in Parents of Children Being Admitted for Febrile Seizures in a Tertiary Hospital in the East Coast of Malaysia

Azizah Othman [1,2], Salmi Abdul Razak [1,2], Ariffin Nasir [1,2], Anis Kausar Ghazali [2,3] and Muhammad Amiro Rasheeq Mohd Radzi [1,2,*]

1   Department of Paediatrics, School of Medical Sciences, Universiti Sains Malaysia, Kubang Kerian 16150, Malaysia; azeezah@usm.my (A.O.); salmikb@usm.my (S.A.R.); ariffinkb@usm.my (A.N.)
2   Hospital Universiti Sains Malaysia, Kubang Kerian 16150, Malaysia; anisyo@usm.my
3   Unit Biostatistics and Research, School of Medical Sciences, Universiti Sains Malaysia, Kubang Kerian 16150, Malaysia
*   Correspondence: mar@usm.my

**Abstract:** Febrile seizures in children are an alarming experience for parents. This study aimed to assess the psychological functioning of parents of children when they were being admitted for treatment of febrile seizures in the hospital, the importance of which is clear, since parents are the primary custodian of their children. This is a cross-sectional study conducted on 110 participants whose child had been admitted for a febrile seizure to Hospital Universiti Sains Malaysia from September 2020 until June 2021. The depression, anxiety, and stress levels were determined based on a validated Bahasa Melayu questionnaire of the Depression Anxiety Stress Scale (DASS-21). In addition, multiple logistic regression was used to determine the associated factors related to the participants' psychological functioning. The mean age of children with febrile seizures were 21 months old, and most children showed features of simple febrile seizures (71.8%). The prevalence of anxiety, stress, and depression were 58.2%, 29%, and 23.6%, respectively. Using multiple logistic regression, child age, family history of febrile seizures, family history of epilepsy, and length of stay in the ward were found to be significantly associated with anxiety when adjusted for other variables. Otherwise, for depression and stress, no significant associated variables were found when adjusted for other variables. Anxiety was highly reported by participants when their children were admitted for febrile seizures. Several factors impacted their anxiety, including the lower the child's age was, participants with no family history of febrile seizures before, and the longer duration of hospital stay. Therefore, further study and intervention on reducing the parent's anxiety could be emphasized in the future.

**Keywords:** febrile seizures; depression; anxiety; stress; parents

## 1. Introduction

Febrile seizures are disturbing neurological experiences among children that account for 1% of emergency visits to the hospital [1]. It is the most frequent seizure type that involves 2–5% of children below five years of age [2,3]. Based on a study done by Worawit Kantamalee et al., it was recorded that one in every 25 children among the general population will experience febrile seizures at least once in their lifetime [4]. In East Asia and Bombay, India, the prevalence of febrile seizures is around 8% to 11% and 1.8% respectively [5–7]. In Malaysia, no published prevalence study had been performed so far [8].

Despite its benign nature, febrile seizures are one of the most common reasons for admission to pediatric emergency departments worldwide [9]. Additionally, witnessing their child undergo febrile seizures is a terrifying experience for many parents [10]. Most

parents thought that their child might die because of a febrile seizure [11]. The situation may also cause severe anxiety and worsening panic conditions in parents who think that their child may either die or suffer from brain damage if they survived [12,13]. The event may then be followed by restrictions on the quality of family life, and parents might suffer anxiety and insecurity for an extended period of time whenever their child subsequently develops a fever [14]. In a study on psychological outcomes in parents of hospitalized children by Robyn Stremler et al., the results showed that 24% of parents achieved scores characteristic of severe anxiety, while parents with symptoms of major depression and significant decisional conflict were 51% and 26%, respectively [15].

Psychological functioning of parents is crucial in maintaining a healthy and balanced society in general. More specific to this study, the psychological functioning of parents is especially important due to the fact that their children are wholly dependent and in the custody of their parents. Therefore, the objective of this study was to assess the psychological functioning of parents of children admitted for febrile seizures. Additionally, this study aimed to assess the relationship between psychological functioning among parents of febrile seizure with their sociodemographic status and clinical characteristics of their child suffering from febrile seizures. Identifying the relevant risk factors that affect their psychological aspects would assist health practitioners in communicating with parents when they face this situation.

## 2. Materials and Methods

### 2.1. Participants

This was a cross-sectional study conducted among participants whose children were admitted for febrile seizures to Hospital Universiti Sains Malaysia over ten months, from September 2020 until June 2021. This study was approved by the Universiti Sains Malaysia Research Ethics Committee (USM/JEPeM/20060321) and carried out in accordance with the Declaration of Helsinki. The following inclusion selection criteria were applied. (1) All participants whose children were of 3 months old to 6 years old, admitted for simple or complex febrile seizure at the pediatric ward. The exclusion criteria include: (1) patients who were subsequently diagnosed or treated with meningitis or encephalitis, (2) underlying developmental delay and medical illness, and (3) participants who had existing mental health illness or been diagnosed and receiving treatment for psychological illness.

### 2.2. Procedures

The data were collected by a self-administered questionnaire with guidance from the researcher. Potential participants were approached directly on the next day of their child being admitted to the pediatric ward. After informed consent, participants completed the assessment measures with the assistance of researchers. Data collection only uses code numbers to ensure confidentiality. The clinical data were retrieved from the children's hospital folder.

The proforma was divided into three sections; participants' sociodemographic characteristics, background data of the child, and participants' psychological health. The data included a list of variables generally associated with psychological distress in the populations. Sociodemographic data included background data of participants, such as age, race, education level, total household income, number of dependencies, employment, and prior knowledge about febrile seizures. The background data of the child included age during admission, the characteristics of febrile seizures, the length of stay, and any family history of febrile seizures. The data also included the presence of any past admission, the duration of fever before the febrile seizures, and the aetiology.

### 2.3. Instruments

The validated Malay version of the Depression, Anxiety, and Stress Scale (DASS-21) was used to assess the participants' psychological functioning. It had 21 items with seven items for each variable. The minimum score for each item was 0, and the maximum score

was 3. The questionnaire had been translated and validated in the Malay language with a Cronbach alpha of 0.74–0.84. The participants were asked to use 4-point scales to rate the extent of which they experienced when witnessing their child having a febrile seizure. The total scores for depression, anxiety, and stress were calculated by summing the scores for the relevant items in DASS-21, and the values were multiplied by two to calculate the final score. Thus, the score for each subscale was 0–42. The participants were categorized as anxious if they squared greater than or equal to 8, depressive if they scored greater than or equal to 10, and stressed if they scored greater than or equal to 15 [16]. The previous study by Rahma Al-Zahrani, entitled "Prevalence of psychological impact on caregivers of hospitilized patients: The forgotten part of the equation", also used DASS-21 as its tool to measure the psychological functioning among caregivers of hospitalized patients. The questionnaire was given to the caretakers who stayed in the hospital and agreed to participate in the study [17].

### 2.4. Data Analysis

The data were analyzed by Statistical Package for the Social Sciences (SPSS) Statistics for Windows (Version 25.0; Armonk, NY, USA, IBM Corp.). Descriptive statistics, including mean and standard deviation, were carried out for sociodemographic data and to determine the level of psychological functioning. Simple and multiple logistic regressions were used in confirmatory analysis to determine the association between the level of psychological functioning among parents of febrile seizure patients with their sociodemographic status and the selected clinical characteristics of children with a febrile seizure. Simple logistic regression was also initially performed on each independent variable to determine any association between the sociodemographic background of the participants and clinical characteristics of their children with febrile seizures with psychological functioning parameters of depression, anxiety, and stress. Those statistically significant variables at the level of $p < 0.25$ were included in the multiple logistic regression analysis. While in the multiple logistic regression analysis, variables were considered significant at the level of $p < 0.05$.

### 3. Results

A total of 110 participants were involved in this study. Their demographic details were presented in Table 1. The results of the data showed that the mean age of the total participants were 33.03 years old with a standard deviation of 5.44. The participants' age ranges from 21 to 47 years old. The majority of the participants were mothers (72.7%). The maximum total income per household recorded in this study was RM 10,000 per month (USD 2381), and the minimum income recorded was RM 450 (USD 107). Predominantly, 99.1% of Malay participants were in the study population ($n = 109$) (Table 1).

The clinical characteristics of children with febrile seizures were summarized in Table 2. The majority of children with febrile seizures were boys (69%). The mean age for children was 21 months, and the children's age ranged from three months to six years old. This study recorded that most of the children studied had simple febrile seizures (71.8%) (Table 2).

The participants reported depression, anxiety, and stress symptoms, which were presented in Table 3.

The associations between all possible variables with depression, anxiety, and stress level were summarized in Table 4. There was a significant association between some of the variables towards depression, anxiety, and stress domain of DASS based on simple logistic regressions towards the presence of the psychological problem. Therefore, the variables with $p < 0.250$ were chosen to proceed with multiple logistic regression. Education level, total family income, type of febrile seizures, and etiology were listed for attention from the depression domain. For anxiety, preceding knowledge, child age, family history of epilepsy, family history of febrile seizures, and length of stay were used. For stress, preceding knowledge was the only significant variable.

**Table 1.** Participants characteristics (*n* = 110).

| Variables | Mean (SD) | *n* (%) |
|---|---|---|
| Participants | | |
| Father | | 30 (27.3) |
| Mother | | 80 (72.7) |
| Race | | |
| Malay | | 109 (99.1) |
| Chinese | | 1 (0.9) |
| Age, years | 33.0 (5.4) | |
| Education level | | |
| Secondary school | | 57 (51.8) |
| College/Graduate | | 53 (48.2) |
| Monthly household income (total monthly income in * thousand RM), Median (IQR) | 2850.0 (3500.0) | |
| Number of dependancy, Median (IQR) | 3.0 (2.0) | |
| Employment | | |
| Government | | 28 (25.4) |
| Private | | 31 (28.2) |
| Self-employed | | 51 (46.4) |
| Preceding knowledge about febrile seizures | | |
| Yes | | 86 (78.2) |
| No | | 24 (21.8) |
| Source of information | | |
| Internet/TV/magazines | | 17 (19.8) |
| Relatives/friends | | 34 (39.5) |
| Hospital (doctor/nurse) | | 35 (40.7) |

* RM1000 is equivalent to about USD 237, Median (IQR), Mean (SD).

Table 5 summarized the associations between all related variables towards depression, anxiety, and stress when adjusted for other variables (using multiple logistic regression). For anxiety, child age ($p < 0.050$), family history of febrile seizures ($p = 0.027$), family history of epilepsy ($p = 0.045$), and length of stay ($p = 0.041$) were found to be significantly associated with anxiety when adjusted for other variables. For children aged 37 to 48 months old, there were 81% lower odds to experience anxiety compared to participants with a child aged 1–12 months old (OR = 0.18 (95%CI = 0.03, 0.99), $p = 0.049$). For participants with no family history of febrile seizures, they were 2.67 times more likely to experience anxiety compared to participants with a positive family history of febrile seizures (OR = 2.67 (95%CI = 1.12, 6.35)). For participants with a family history of epilepsy, there was 84% lower odds to experience anxiety compared to participants with no history of epilepsy (OR = 0.16 (95%CI = 0.03, 0.96)). For the participants whose children have stayed three and more days in hospital due to recent febrile seizure, there were 2.75 times odds to experience anxiety compared to participants with a stay of one to two days (OR = 2.75 (95%CI = 1.04, 7.23)) only. For depression and stress, no significant associated variables were found when adjusted for other variables.

**Table 2.** Clinical characteristic of children with febrile seizures.

| Variables | *n* (%) |
|---|---|
| Gender | |
| Boy | 76 (69.1) |
| Girl | 34 (30.9) |
| Age during admission (month) | |
| Mean (SD) | 21.54 (15.15) |
| 1–12 | 31 (28.2) |

**Table 2.** *Cont.*

| Variables | *n* (%) |
|---|---|
| 13–36 | 62 (56.4) |
| 37–48 | 10 (9.1) |
| >49 | 7 (6.4) |
| Characteristics of FS | |
| Simple FS | 79 (71.8) |
| Complex FS | 31 (28.2) |
| Duration of fever before onset of seizure (Hours) | |
| <24 | 77 (70.0) |
| 24 h or more | 33 (30.0) |
| Past admission for other reason | |
| Yes | 45 (40.9) |
| No | 65 (59.1) |
| Family history of febrile seizures | |
| Yes | 55 (50.0) |
| No | 55 (50.0) |
| Family history of epilepsy | |
| Yes | 8 (7.3) |
| No | 102 (92.7) |
| Length of present admission (days) | |
| 1–2 | 75 (68.2) |
| 3 days and more | 35 (31.8) |
| Etiology | |
| Acute tonsillitis/pharyngitis | 61 (55.5) |
| Viral fever | 21 (19.1) |
| Bronchopneumonia | 4 (3.6) |
| Others (gastroenteritis, ear infections, UTI) | 24 (21.8) |

**Table 3.** Level of psychological functioning (DAS).

| DASS-21 | Normal | Abnormal Scores |
|---|---|---|
| Depression score | 84 (76.4%) | 26 (23.6%) |
| Anxiety score | 46 (41.8%) | 64 (58.2%) |
| Stress score | 88 (80%) | 22 (20.0%) |

**Table 4.** Factors associated with depression, anxiety, and stress.

| Independent Variables | Crude OR (95% CI) | *p*-Value | Crude OR (95% CI) | *p*-Value | Crude OR (95% CI) | *p*-Value |
|---|---|---|---|---|---|---|
| Participant characteristics | | | | | | |
| | (Depression) | | (Anxiety) | | (Stress) | |
| Age (year) | 1.00 (0.92, 1.09) | 0.97 | 1.00 (0.93, 1.07) | 0.993 | 0.98 (0.90, 1.07) | 0.64 |
| Education level | | | | | | |
| Secondary school | 1 | | 1 | | 1 | |
| College/graduate | 0.39 (0.15, 0.98) | 0.046 | 1.03 (0.48, 2.19) | 0.950 | 0.69 (0.27, 1.79) | 0.447 |
| Monthly income of household | 1.00 (0.92, 1.09) | 0.435 | 1.00 (1.00, 1.00) | 0.59 | 1.00 (1.00, 1.00) | 0.64 |
| Preceding knowledge about febrile seizures | | | | | | |
| Yes | 1 | | 1 | | 1 | |
| No | 1.1 (0.39, 3.15) | 0.859 | 0.50 (0.19, 1.32) | 0.160 | 0.39 (0.14, 1.08) | 0.071 |

**Table 4.** *Cont.*

| Independent Variables | Crude OR (95% CI) | *p*-Value | Crude OR (95% CI) | *p*-Value | Crude OR (95% CI) | *p*-Value |
|---|---|---|---|---|---|---|
| Number of dependants | | | | | | |
| (1–2, | 1 | | 1 | | 1 | |
| 3, or more) | 0.65 (0.26, 1.59) | 0.343 | 1.20 (0.54, 2.66) | 0.652 | 0.71 (0.27, 1.85) | 0.484 |
| Employment | 1 | | | | | |
| Government | 2.89 | | 1 | | 1 | |
| Private | (0.78, 10.47) | 0.113 | 0.58 (0.20, 1.67) | 0.308 | 0.88 (0.25, 3.13) | 0.843 |
| Self-employed | 1.85 (0.53, 3.60) | 0.333 | 0.58 (0.22, 1.52) | 0.264 | 0.89 (0.29, 2.78) | 0.847 |
| Children characteristics | | | | | | |
| Age during admission (month) | | | | | | |
| (1–12) | 1 | | 1 | | 1 | |
| (13–36) | 1.57 (0.55, 4.50) | 0.398 | 0.61 (0.24, 1.53) | 0.289 | 0.59 (0.22, 1.59) | 0.296 |
| (37–48) | 1.04 (0.17, 6.22) | 0.964 | 0.18 (0.04, 0.83) | 0.029 | 0.27 (0.03, 2.47) | 0.247 |
| (49 and more) | 0.69 (0.07, 6.91) | 0.756 | 0.16 (0.03, 1.00) | 0.050 | 0.00 (0.00, 0.00) | >0.950 |
| Characteristics of FS | | | | | | |
| Simple FS | 1 | | 1 | | 1 | |
| Complex FS | 2.35 (0.93, 5.92) | 0.071 | 0.99 (0.43, 2.31) | >0.950 | 1.24 (0.45, 3.43) | 0.672 |
| Family history of febrile seizures | | | | | | |
| No | 1 | | 1 | | 1 | |
| Yes | 0.67 (0.27, 1.62) | 0.371 | 2.13 (0.99, 4.62) | 0.055 | 0.80 (0.31, 2.03) | 0.634 |
| Family history of epilepsy | | | | | | |
| No | 1 | | 1 | | 1 | |
| Yes | 0.00 (0.00, 0.00) | >0.950 | 0.22 (0.04, 1.l2) | 0.068 | 0.00 (0.00, 0.00) | >0.950 |
| Past admission | | | | | | |
| Yes | 1 | | 1 | | 1 | |
| No | 0.93 (0.38, 2.26) | 0.868 | 1.40 (0.65, 3.02) | 0.392 | 1.63 (0.60, 4.39) | 0.335 |
| Length of present admission (days) | | | | | | |
| One to two | 1 | | 1 | | 1 | |
| Three or more | 1.48 (0.59, 3.69) | 0.407 | 2.31 (0.98, 5.46) | 0.057 | 1.00 (0.37, 2.73) | >0.950 |
| Duration of fever before admission (Hours) | | | | | | |
| <24 | 1 | | 1 | | 1 | |
| 24 and more | 1.32 (0.52, 3.38) | 0.557 | 0.97 (0.42, 2.20) | 0.933 | 1.44 (0.54, 3.85) | 0.468 |
| Etiology | | | | | | |
| Acute tonsillitis/pharyngitis | 1 | | 1 | | 1 | |
| Viral fever | 0.27 (0.06, 1.30) | 0.103 | 0.87 (0.32, 2.36) | 0.777 | 0.51 (0.13, 1.98) | 0.331 |
| Bronchopneumonia | 0.00 (0.00, 0.00) | >0.950 | 0.65 (0.09, 4.92) | 0.675 | 0.00 (0.00, 0.00) | >0.950 |
| Others | 1.07 (0.38, 3.03) | >0.950 | 0.77 (0.30, 1.99) | 0.585 | 0.61 (0.18, 2.08) | 0.433 |

**Table 5.** Summary of significant factors associated with depression, anxiety, and stress.

| DASS Subscale | Factor | Adjusted OR (95% CI) | *p*-Value |
|---|---|---|---|
| Depression | | - | - |
| Anxiety | | | |
| | Child age during admission (month) | | |
| | 1–12 | 1 | |
| | 13–36 | 0.64 (0.24,1.71) | 0.370 |
| | 37–48 | 0.81 (0.03,0.99) | 0.049 |
| | 49 and more | 0.18 (0.03,1.16) | 0.071 |
| | Family history of febrile seizures | | |
| | Yes | 1 | |
| | No | 2.67 (1.12,6.35) | 0.027 |
| | Family history of epilepsy | | |
| | No | 1 | |
| | Yes | 0.16 (0.03, 0.96) | 0.045 |
| | Length of present admission (days) | | |
| | One to two | 1 | |
| | Three or more | 2.75 (1.04, 7.23) | 0.041 |
| Stress | - | - | - |

Most of the assumption met for anxiety model. Anxiety: Classification table = 67.3%, Hosemer and Lemeshow test = *p* 0.889, ROC = 0.737 (0.644, 0.831).

## 4. Discussion

In ten months of data collection, our study showed that 26 out of 110 (23.6%) enrolled respondents reported depressive symptoms, 64 out of 110 (58.2%) had anxiety, and 22 out of 110 (20%) enrolled respondents had stress. Generally, our study showed a considerable amount of anxiety, depression, and stress among parents whose child had been admitted for febrile seizures.

Based on our study, anxiety turned out to be higher among parents of children with febrile seizures compared to the depression and stress component. This can possibly be explained according to the nature of the illness. Even though febrile seizures can be categorized as benign conditions, the devastating effect of the event might be too much for some parents. To observe their child having seizures is a tense experience for many parents [10]. Based on a study performed among epilepsy patients, it was found out that there were increased rates of difficulties with depression, anxiety, and stress when compared to the previous research on the impact of having children with neurological or neurodevelopmental conditions [18]. In contrast to that, they also found that the components of depression and stress were likely to be at risk for the parents of children with epilepsy when compared to the parents of children with neurodisability, whereby the symptoms scores for both scales were higher [18]. This suggests that the parents of patients with epilepsy may have a specific tendency to develop depression when compared to the parents with febrile seizure patients, where they are more prone to be higher in anxiety score. The possible reason for this is due to the increased risk for depression in parents, especially mothers of children with epilepsy, because of shared genetic risk for the two disorders. In a previous study, it was shown that there was a bidirectional relationship between epilepsy and depression, and this could be due to similar underlying pathophysiological mechanisms that both lower seizure tendency or threshold and increase the risk for psychiatric conditions [19]. It is not child epilepsy-related factors alone that can contribute to the depression episode of the parents. Thus, these findings suggest that the risk could be associated with multiple factors with a shared background

factor, resulting in an increased tendency for epilepsy in the child and depression in parents, especially the mothers. However, the nature of epilepsy is that it is unpredictable, and the tendency to accidentally fall can cause further risk of injury, which may possibly contribute to cognitive impairment and increased mental health problems [18].

Thus far, it was found that the anxiety component was higher among parents of patients with febrile seizures when compared to epilepsy parents who had a higher depression component. One of the reasons is that epilepsy is an ongoing, potentially life-long condition, whereas febrile seizures are benign and less likely to recur. Besides, this condition is possibly due to the nature of febrile seizures itself. For the parents who are witnessing it for the first time, it might be an uncomfortable and alarming experience for them [20–22]. Due to the unpleasant condition of febrile seizures that they witnessed happening on their child, their parental worries are therefore understandable. In our study, out of 110 parents, 64 of them (58.2%) showed symptoms of anxiety based on the DASS-21 scale system. The results reflect the true present emotion, as well as the state anxiety (not necessarily trait) of parents that are experiencing the tense event regarding their child. This finding could be explained by the study performed by Azizi et al. [8], who found that 83 (70.9%) out of 117 parents interviewed said that they were afraid that their child was likely to pass away or die as a final result of the febrile seizures. Hughes et al. also showed similar findings when they found that 35 (70%) out of 50 parents were also concerned and questioned their thought that their child was likely to die or was dying or dead [21]. Additionally, in a study performed by Metcalfe, they found that parents who witnessed their child having a seizure was frightened, perplexed, and confused, and 30% of them thought their child was dead or dying [20]. Their study showed different findings compared to ours, which could be possible due to the different approaches to how the questions were delivered. For example, in the study performed by Metcalfe mentioned above, they had a lower percentage of parents who were in a state of anxiety than those performed by Hughes et al. because they were using open-ended questions instead of direct questioning. Thus, in their study, some parents might have hidden their fear. In Balslev's study, only 34% expressed fear when specifically asked about it, and about 54% of parents volunteered their fear [22]. In a study performed in Japan, 44.5% of the parents volunteered the fear of witnessing their child having febrile seizures. Based on the above results, it can be concluded that anxiety and fear are in close relation with each other and interrelated. In a study performed by Maya et al., there is a significant positive relationship between anxiety and fear of negative evaluation. It means that the fear of negativism and the state-trait of anxiety are highly correlated [23].

Interestingly, in our study, the anxiety part of parents was found to have an association with the age of their children who are having febrile seizures. The older the child is, the lesser the tendency for the parents to experience anxiety. This could be due to the fact that the younger the child is, the more likely that the parents might think some unfortunate events will occur to their child. In the past literature, there were contradictory reports on whether the child's age is relevant to the parent's psychological functioning in the case of illness. In a study performed by Tiedeman, there were no difference in the intensity of anxiety experienced by parents of younger and older children [24]. While, in another study performed by Krywda-Rybska et al., it was found that the parents of younger children were found to be more stressed and anxious [25]. This finding is consistent with a previous study, which showed that, when the child's age increases, this will then reduce the stress to its parents [26].

In our study, participants with no family history of febrile seizures are at a greater risk for experiencing anxiety. In febrile seizures, they may only occur once, and parents usually are not experienced enough to tell whether their child is stable or not. The parents' perception was supported by the study performed by Panteliadis et al., where children with positive family history of febrile seizures were at greater risk for recurrence [27]. This result is similar to a study performed by Louthrenoo et al., where they found that family history of febrile seizures was a significant risk factor for recurrent febrile seizures [4]. Thus,

the recurrence of febrile seizures will eventually influence the evolution of participants anxiety [28]. However, the study performed by Bethune et al. found that the children with a known family history of febrile seizures had 24% fewer physician visits [29]. Furthermore, the first time that a febrile seizure occurs might not be anticipated and, hence, the situation might result in great anxiety for the parents.

This study also found that participants whose children had stayed more than two days in the hospital were more prone to be more anxious than participants whose children had stayed less than two days. This may be due to the fact that, when parents stay longer in the hospital, they face the uncertainty regarding their child's condition. Children who experienced febrile seizures are usually advised for hospital admission for observations. In the case of complex febrile seizures, it was highly associated with recurrence. A previously performed study also showed that those patients with three or more seizures in 24 h have an increased risk for further seizure [30]. Thus, with the higher number of seizures and the need for observation in the ward, it could indirectly affect the anxiety symptoms of the parents themselves. In a recent study performed by Jacobs et al., increasing anxiety levels were observed using the quantitative manner of the State-Trait Anxiety Inventory (STAI). Based on their results, the anxiety scores were high in both conditions of parents whose children suffered from a first unprovoked febrile seizure and first febrile seizure [28].

The study questionnaire was given to the parents on the next day of admission. While the child's condition might be more stable, parent's emotions might have improved. However, our study findings still indicate a high level of anxiety among parents of febrile seizure patients. The study performed by Franck et al. showed that parental anxiety, depression, and stress increased shortly after their child's admission and remained high at discharge. This was due to some parents being more influenced by emotional factors, coping styles, the state of uncertainty, and the level of optimism in parents of hospitalized children [31].

This study had several limitations. First, the participants involved were from one local center. Thus, the findings were less generalizable to the entire population. For better associations and representations, we suggest that future research may involve a larger number of samples. The self-reported DASS-21 scales used in this study are just a screening tool. They may not reveal valid and actual nature of depression, anxiety, and stress, and they also may not always be consistent compared to an assessment by a mental health professional. Furthermore, this study was conducted during the beginning of the COVID-19 pandemic, of which the maximum impact and effect of the pandemic may not have been established. Finally, as this was a cross-sectional study, it does not reveal cause-and-effect relationships to be studied.

To the best of our knowledge, there were no studies yet performed in Malaysia to assess the psychological functioning in parents of children with febrile seizures, in particular, their depressive, anxiety, and stress symptoms using the DASS-21 scale questionnaire. We agree that our results can be helpful in designing strategies for early identification of mental health disorders, as well as psychological and other interventions, leading to mental health promotion and wellbeing in the population. Further study and intervention on reducing the parent's anxiety could be emphasized more in the future.

## 5. Conclusions

In conclusion, participants consistently reported anxiety when their children were admitted for febrile seizures. Several factors impacted their anxiety, including the child's age, lack of family history of febrile seizures, and the duration of hospital stay. Therefore, it is essential to recognize the issue early so that interventions, such as counseling, can be offered.

**Author Contributions:** Conceptualization, A.O. and M.A.R.M.R.; Formal analysis, A.O. and M.A.R.M.R.; Methodology, S.A.R., A.N. and A.K.G.; Supervision, A.O., S.A.R., A.N. and A.K.G.; Validation, S.A.R.; Writing—original draft, M.A.R.M.R.; Writing—review and editing, A.O. and M.A.R.M.R. All authors have read and agreed to the published version of the manuscript.

**Funding:** This research did not receive any specific grant from funding agencies in the public, commercial, or not-for-profit sectors.

**Institutional Review Board Statement:** This study was approved by the Institutional Review Board of the Human Research Ethics Committee, Universiti Sains Malaysia Research Ethics Committee (JePeM code: USM/JEPeM 20060321), and carried out in accordance with the Declaration of Helsinki.

**Informed Consent Statement:** Informed consent was obtained from all subjects involved in the study.

**Data Availability Statement:** The data presented in this study are available upon request from the corresponding author.

**Acknowledgments:** A special thanks is dedicated to the research ethics committee of Hospital Universiti Sains Malaysia, fellow lecturers, colleagues, supporting staff, participants, family, and team of authors whom journals used as references.

**Conflicts of Interest:** The authors declare no conflict of interest. The funders had no role in the design of the study, in the collection, analyses, or interpretation of data, in the writing of the manuscript, or in the decision to publish the results.

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
