# Peer review of "Depressive, Anxiety, and Stress Symptoms in Parents of Children Being Admitted for Febrile Seizures in a Tertiary Hospital in the East Coast of Malaysia"

_ejihpe, doi:10.3390/ejihpe13060077_

Round 1

Reviewer 1 Report

The article is well done.

Author Response

We would like to express our sincere appreciation for the positive feedback and kind words regarding our article. Thank you for your time and valuable input.

Reviewer 2 Report

Studying the psychological functioning of parents of children when they were being admitted for treatment of febrile seizures is with significant implications for medical practice. The text is clear and easy to read. The manuscript has an excellent methodical description. The overall paper is organized and well-written. The methods, the overall study design, and the statistical analysis are clearly described. The discussions Section presents other research findings. The literature reviews are insightful and informative. The tables are well-presented and easy to read and understand. The presented aspects sufficiently support the conclusions.  I congratulate all the authors for their efforts.

Author Response

We would like to express our sincere gratitude to the reviewers for their valuable comments and feedback on our manuscript. Their insights and suggestions have been instrumental in enhancing the quality and significance of our study.

Studying the psychological functioning of parents of children during their admission for febrile seizure treatment holds significant implications for medical practice, and we are grateful for the reviewers' recognition of this. We are pleased to hear that the text is clear and easy to read, as we have made efforts to ensure the accessibility of our research.

Furthermore, we appreciate the positive feedback regarding the methodical description and organization of the overall paper. It is encouraging to know that our methods, study design, and statistical analysis were clearly communicated and well-presented.

The reviewers' acknowledgment of the informative and insightful literature reviews, as well as the well-presented and understandable tables, is greatly appreciated. We strived to provide a comprehensive overview of relevant research findings and to present our data in a visually accessible manner.

We are grateful for the reviewers' time and expertise, which have undoubtedly contributed to the advancement of our study.

Reviewer 3 Report

This study is highly significant in the field of parents’ mental health for young child health in Malaysia. This is an important topic that warrants greater attention in focusing parent’s anxiety for effective intervention. 

  • In this study, there is a lack of previous research on the relationship between parental mental health and child physical health, and it is necessary to suggest what role it plays in Malaysia. I am wondering if there is information about children being admitted for febrile seizures in hospitals. 

  • In the research methodology, the study clearly indicates how all variables used in this study were measured.

  • This study also discusses what the limitations of this study are.

  • Discussion section needs to contain lots of rich arguments, which is very worthwhile. The points would read better with a clear that what are the summaries of the main findings, what research implications and suggestions for future research are discussed, and what kinds of infant development implications can be made.
